# Development and Experimental Validation of a Novel Prognostic Signature for Gastric Cancer

**DOI:** 10.3390/cancers15051610

**Published:** 2023-03-05

**Authors:** Chengcheng Liu, Yuying Huo, Yansong Zhang, Fumei Yin, Taoyu Chen, Zhenyi Wang, Juntao Gao, Peng Jin, Xiangyu Li, Minglei Shi, Michael Q. Zhang

**Affiliations:** 1School of Software Engineering, Beijing Jiaotong University, Beijing 100044, China; 2MOE Key Laboratory of Bioinformatics, Bioinformatics Division and Center for Synthetic & Systems Biology, BNRist, Tsinghua University, Beijing 100084, China; 3School of Medicine, Tsinghua University, Beijing 100084, China; 4School of Life Sciences, Peking University, Beijing 100871, China; 5Medical School of Chinese PLA, Beijing 100853, China; 6Department of Bioinformatics, School of Basic Medicine, Peking University Health Center, Beijing 100191, China; 7Department of Automation, Tsinghua University, Beijing 100084, China; 8Senior Department of Gastroenterology, First Medical Center of Chinese PLA General Hospital, Beijing 100036, China; 9Department of Biological Sciences, Center for Systems Biology, University of Texas at Dallas, Richardson, TX 75080, USA

**Keywords:** gastric cancer, prognostic signature, tumor microenvironment, oncogenic mutation, clinical outcomes, machine learning

## Abstract

**Simple Summary:**

Gastric cancer (GC) accounts for a considerable amount of morbidity and mortality worldwide. This study developed and experimentally validated a prognostic risk gene signature (PRGS). This is a stable and robust signature for assessing the prognosis of gastric cancer. We performed multiple analyses through consensus clustering and binary classification to assess the robustness of the PRGS in other independent datasets. Additionally, this PRGS exhibited a superior accuracy compared to most traditional clinical markers, including molecular features, and other published signatures. Besides, we also detected the tumor purity, immune cell infiltration, and oncogenic mutation status of high- and low-PRGS groups.

**Abstract:**

Background: Gastric cancer is a malignant tumor with high morbidity and mortality. Therefore, the accurate recognition of prognostic molecular markers is the key to improving treatment efficacy and prognosis. Methods: In this study, we developed a stable and robust signature through a series of processes using machine-learning approaches. This PRGS was further experimentally validated in clinical samples and a gastric cancer cell line. Results: The PRGS is an independent risk factor for overall survival that performs reliably and has a robust utility. Notably, PRGS proteins promote cancer cell proliferation by regulating the cell cycle. Besides, the high-risk group displayed a lower tumor purity, higher immune cell infiltration, and lower oncogenic mutation than the low-PRGS group. Conclusions: This PRGS could be a powerful and robust tool to improve clinical outcomes for individual gastric cancer patients.

## 1. Introduction

Gastric cancer (GC) is a leading cause of cancer morbidity and mortality worldwide [1]. According to GLOBOCAN 2020, there were about 1,089,103 new cases of gastric cancer patients (5.6% of the total cancer burden) [1]. Disease progression and a lack of effective treatment cause most of the mortality [2]. Therefore, preventing “high-risk” GC is the key to improving clinical outcomes. The tumor, node, metastasis (TNM) classification and the American Joint Committee on Cancer (AJCC) classification [3] are commonly used methods to assess the risk and treatment demand for patients in the clinical setting. Nevertheless, due to the limitation of the current grading system, it cannot provide the best clinical treatment for patients. For example, in the clinic, the decision of adjuvant chemotherapy (ACT) is mainly dependent on the clinical–pathological stage rather than molecular biological characteristics [4]. This approach is insufficient and may result in latent overtreatment or undertreatment. Hence, in the era of individualized treatment, it is imperative to identify effective biomarkers to optimize the prognosis of GC.

The ideal biomarker should have a consistent expression within and between tumor tissues to perform stably among all patients. Hence, a multigene signature may be an effective approach for addressing this heterogeneity. To date, there are three categories of clinically important GC markers, CEA, CA19-9, and CA72-4, with positive rates of 21.1%, 27.8%, and 30.0%, respectively [5]. However, they are present in a limited number of patients with GC, and the sensitivity and specificity of these biomarkers are not sufficient. With the development of bioinformatics technology, many prognostic biomarkers have been published [6,7,8]. Unfortunately, most of the identified biomarkers failed in the validation.

GC is a heterogeneous malignant disease. Histologically, the human gastric mucosa can be divided into three zones, i.e., the cardiac zone, the fundus/corpus zone, and the pyloric zone [9,10]. These zones differ vastly in their histology, regeneration rates, and profiles [11]. In addition, there are different morbidity and mortality rates among GC in these three zones. It is believed that GC patients with the GC localized in the cardiac zone have the worst prognosis. As the patient’s age progresses, the location of the gastric cancer moves upward, and the incidence of it occurring in the cardiac zone increases [12]. The composition of cell types within them is also discrepant. For instance, the quantities of gastric parietal cells are considerably different in these three zones. Namely, parietal cells account for 25%, 50–100%, and 0–1% of all cells in the cardiac zone, the fundus/corpus zone, and the pyloric zone, respectively [12]. Of note, most researchers who investigate gastric cancer biomarkers have ignored the differences between these three zones and analyzed them as if all gastric cancers are of one single area. Hence, it is essential to construct gene co-expression networks and select prognostic-related genes separately for GC data in these three zones.

In this work, we attempted to computationally develop and experimentally validate a prognostic risk gene signature with 1226 GC patients from three independent public datasets (Appendix A), a gastric cancer cell line, and several clinical samples to assess the prognosis, tumor growth, and molecular characterization of GC. A multi-step procedure of machine-learning approaches was performed to develop and cross-validate the prognostic risk gene signature (PRGS) model based on the co-expression networks of GC in the cardiac zone, the fundus/corpus zone, and the pyloric zone. This PRGS may help optimize precision treatment and further improve the clinical outcomes of GC patients.

## 2. Materials and Methods

### 2.1. Data Acquisition and Processing

In this study, we used three independent public datasets, including 1226 GC patients obtained from the UCSC Toil Recompute Compendium of The Cancer Genome Atlas TARGET and Genotype Tissue Expression project datasets (TCGA target GTEx, primary_site = stomach) [13] and the Gene Expression Omnibus (GEO) (Appendix A). These datasets (TCGA target GTEx, GSE66229, and GSE15459), which encompass complete overall survival (OS) information, were used. Among these, we converted the RNA-seq raw read count from the TCGA target GTEx database to transcripts per kilobase million (TPM) and then log2-transformed. And the data has been removed batch effects among these patients [14]. We retrieved GSE15459 and GSE66229 from the Affymetrix^®^ GPL570 platform (Human Genome U133 Plus 2.0 Array). We selected the most highly expressed probe for each gene to ensure reliable results in consensus clustering and we reserved all probes of each gene to ensure accurate results in binary classification.

The ATAC-seq and somatic variation data were obtained from the database of the Genomic Data Commons Data Portal (https://portal.gdc.cancer.gov/, accessed on 1 February 2022).

### 2.2. Human Tissue Specimens

In this study, the collection of gastric tissues was approved by the Department of Gastroenterology, Seventh Medical Center of Chinese PLA General Hospital, Beijing, China, on 1 June 2022. Overall, one advanced GC sample, an early GC sample, and a normal gastric sample were collected. All patients provided written informed consent, and the ethics committee of the PLA General Hospital also approved our research.

### 2.3. Differentially Expressed Analysis and Weighted Correlation Network Analysis

The DEGs between cancer and normal samples were detected by edgeR [15]. The genes with an absolute log2 (fold change) ≥ 1.5 were considered to be significant differentially expressed genes (DEGs) between tumor and normal tissues. The volcano plots of upregulated or downregulated genes were generated by the ggplot2 R 4.0 package. The Venn diagram was plotted by the VennDiagram R 4.0 package.

The co-expression gene networks were constructed by the WGCNA package [16]. To recognize modules of significantly correlated clusters in the cardiac zone, the fundus/corpus zone, and the pyloric zone, the module that displayed the highest correlation was selected.

### 2.4. Construction of the Co-Expression Network

The brown, purple, and salmon modules of the result of WGCNA were used for the edges, signifying the correlations in the cardiac zone, the fundus/corpus zone, and the pyloric zone, respectively. The filter criterion of a weight value was set to greater than 0.02.

A total of 25,350, 1819, and 706 edges and 308, 69, and 54 nodes correlated with these zones were separately obtained and processed in Cytoscape 3.8.1 [17]. To construct the network and select the hub genes, the Cytoscape software was used with the CytoHubba method [18].

### 2.5. PRGS Signature Generation

LASSO is a regularization and dimensionality reduction technique combined with Cox models, which can be applied in biomarker screening [19]. To identify the hub genes, the top 25 genes of the co-expression network from the cardiac zone, the fundus/corpus zone, and the pyloric zone by Cytoscape were inputted into the LASSO–Cox regression [19]. To ensure the stability of the gene and the model, this procedure was repeated 1000 times. Then, the regression coefficients of each gene were calculated (Appendix A). Ultimately, genes with positive regression coefficients were selected. Four genes (APOB, VCAN, ABCA6, and CTSF) whose *p*-values in the Kaplan–Meier analysis were also < 0.05 were identified to generate the PRGS model. The PRGS risk score of GC patients was calculated by the formula: PRGS score = $\overset{4}{\underset{k=1}{\sum }}\left(\begin{array}{c} 4\\ 1 \end{array}\right)$β_k_ × RNA_k_.

In this formula, k means the genes in the PRGS—we used k = 1, 2, 3, 4 to index the genes in the PRGS—the β value was the multivariate Cox regression coefficient; and RNAk means the expression level of gene k of each patient.

### 2.6. Chromatin Accessibility Analysis

The peak regions on chromosomes were shown by the R package chromosome locator. Using the R-packaged ChIPseeker, the alignment can be mapped to peaks in the TSS region to build a signature matrix. Peak position annotation and motif enrichment analyses were performed using HOMER (V4.10). In the range of ±1 kb around the TSS, the peak overlapping the gene initiator was considered as the peak of gene regulation.

### 2.7. Consensus Clustering

This process was accomplished through the ConsensusClusterPlus package [20]. We subsampled 80% of the samples, and then used the k-means algorithm to divide each subsample into k (k = 9) according to the Euclidean distance. This procedure was repeated 1000 times. Finally, using the PAC with the smallest k value, the optimal cluster (k = 2) was derived.

### 2.8. Binary Classification

(1)We pre-processed the GSE66229 data and selected the top 5 genes from the three generated co-expression networks of the three zones’ expression spectrum matrices, then divided the data into normal and tumor samples. We also generated the GSE66229 data and generated PRGS, CEA [21], and GCscore [6] expression spectrum matrices.(2)We performed 5-fold cross validation using the logistic regression classifier (LR) and the random forest classifier (RF) on the GSE66229 and computed the ROC curve for each fold. We then calculated the means of every fitting curve to generate the plot. We then divided the data into five subsets based on the sample tags “Tumor/Normal”, “Stage 1/Normal”, “Stage 2/Normal”, “Stage 3/Normal”, and “Stage 4/Normal”. Details of the data are shown in Appendix A.

The parameters were set as follows: RF, max_depth = 5, n_estimators = 5, random_state = 123; LR, solver = “liblinear”, penalty = “l2”, C = 1.0. The rest were set as default. We trained and validated each fold and calculated the FPR, TPR, and AUC. We fitted each result into the ROC curve using np.interp and took the means of all fitting curves to craft the figure.

### 2.9. Haematoxylin–Eosin (HE) and Immunohistochemistry (IHC)

For HE staining, a 4% paraformaldehyde solution was used to fix GC tissues. After 24 h, the GC tissues were dewaxed in xylene, dehydrated in ethanol, and subjected to hematoxylin staining (5 min), then dehydrated in eosin solution (10 s), dehydrated in graded alcohol, removed in xylene, and sealed with neutral glue. They were observed and photographed with a microscope (Olympus, Tokyo, Japan).

For IHC, 5% bovine serum was used to incubate sections. Then, they were mixed with primary antibodies (anti-APOB, 1:100; anti-VCAN, 1:100; anti-ABCA6, 1:200; and anti-CTSF, 1:200, Abcam, UK) and a secondary antibody (1:800, Abcam, Cambridge, UK). Then, they were stained with a DAB kit and photographed with an optical microscope (Olympus, Japan).

### 2.10. Cell Culture, Transfection, and Immunostaining

GES-1 gastric cancer cells (a gift from Prof. Jun Qin) were cultured with 5% CO_2_ at 37 °C in Dulbecco’s modified Eagle’s medium (DMEM) (Thermo Fisher, Waltham, MA, USA, C11995500BT) with 20% FBS and 100 mg/mL penicillin/streptomycin.

For the siRNA treatment, Lipofectamine™ 3000 Transfection Reagent (Thermo Fisher, L3000001) was used to transfect cells using a standard procedure.

For immunostaining, poly-l-lysine-coated coverslips were used to culture cells for 72 h to perform the siRNA transfection; 4% paraformaldehyde was used to fix the cells for 10 min, and then the cells were washed with PBSTr buffer (PBS plus 0.1% Triton X-100, Sigma-Aldrich, St. Louis, MO, USA T8787). The cells were incubated with the anti-phosphorylated histone 3 (pH3) antibody overnight at 4 °C. The cells were incubated with a secondary antibody at room temperature and washed with PBSTr buffer. Then, 0.01 mg/mL DAPI and the Vectashield Antifade Mounting Medium (Vector Laboratories, San Francisco, CA, USA H-1200) were used to incubate and mount the cells, respectively.

### 2.11. Flow Cytometric Analysis

For flow cytometric analysis of the cell cycle with propidium iodide (PI) staining, a standard procedure was used [22].

### 2.12. Quantitative Real-Time PCR

The Eastep Super Total RNA Extraction Kit (Promega, Madison, WI, USA LS1040) was used to extract the total RNA of siRNA-treated GES-1 cells. The Eastep RT Master Mix Kit (Promega, LS2050) was used to synthesize the cDNA. An Applied Biosystems 7500 real-time PCR system (Thermo Fisher) was used to perform real-time PCRs with the PowerUp SYBR Green Master Mix (Thermo Fisher, A25776). The comparative CT method and Graphpad Prism 8 (GraphPad Software, La Jolla, CA, USA) were used to analyze the data. All experiments were repeated three times.

### 2.13. Cell Infiltration Estimation

ESTIMATE [23] and CIBERSORT [24] were used to evaluate immune infiltrates. The immune scores, stromal scores, and tumor purity were calculated by the ESTIMATE algorithm. CIBERSORT [24] was used to analyze the levels of infiltrating immune and stromal cells.

### 2.14. Tumor Mutation Status Analysis

Maftools was used to calculate significantly mutated genes (*p* < 0.05) between the low- and high-PRGS groups [25]. A one-sided *z*-test and two-sided Chi-square test were used to calculate the statistical test for the proportion of mutations (*p* < 0.05).

### 2.15. Functional Enrichment Analysis

A functional enrichment analysis was performed on DEGs and peak-related genes. The possible peak-related genes of GO/KEGG enrichment were used to analyze the ClusterProfiler package in R [26].

Gene ontology (GO) and the Kyoto Encyclopedia of Genes and Genomes (KEGG) database were used to annotate the tumor-related pathways. The gene set variation (GSVA) method [27] was used to enrich the pathways. We used the enrichment score of the GSVA to obtain the expression pathway of the PRGS.

## 3. Results

### 3.1. Construction of the Gene Modules

The workflow of this work is shown in Figure 1.

First, we identified the differentially expressed genes (DEGs) in the TCGA target GTEx with the edgeR package independently to screen DEGs in normal samples and GC samples from the cardiac zone, the fundus/corpus zone, and the pyloric zone. For GC in the cardiac zone, 1498 upregulated and 883 downregulated DEGs were identified. A total of 1661 upregulated and 826 downregulated DEGs were found between normal and GC samples from the fundus/corpus zone. There were 1542 upregulated and 889 downregulated DEGs for GC in the pyloric zone compared with normal samples (Appendix A). The distributions of DEGs are shown by volcano plots (Figure 2A). There were 1553 DEGs shared among the GC samples from the three zones (Figure 2B). We adopted GO and KEGG enrichment analysis methods to investigate the annotation of the DEGs. The DEGs of GC samples from the cardiac zone were mainly enriched in sensory perception (GO:0050907), the collagen-containing extracellular matrix (GO:0062023), the sarcomere (GO:0030017), and olfactory transduction. Sensory perception (GO:0050907), the ion channel complex (GO:0034702), olfactory receptor activity (GO:0004984), and olfactory transduction were detected in GC samples from the fundus/corpus zone. In addition, the GC samples from the pyloric zone were highly associated with digestion (GO:0007586), the collagen-containing extracellular matrix (GO:0062023), receptor–ligand activity (GO:0048018), and neuroactive ligand–receptor interactions (Appendix A).

To further identify the gene modules related to GC in the cardiac zone, the fundus/corpus zone, and the pyloric zone, the WGCNA method [16] was applied. We assured a scale-free network (soft threshold = 3) with a high scale independence and a low mean connectivity (near 0) (Appendix A). DEGs in the GC samples from the cardiac zone, the fundus/corpus zone, and the pyloric zone were respectively divided into 22 modules by a cluster analysis (Figure 2C). The brown module related to GC in the pyloric zone was the most significant (cor = 0.76, *p* = 2 × 10^−112^). For GC in the cardiac zone and the fundus/corpus zone, the purple and salmon modules were chosen according to the correlation (cor = 0.22, 0.33; *p* = 3 × 10^−8^, 1 × 10^−16^).

We selected the brown, purple, and salmon modules for the edges, representing the correlations in GC in the cardiac zone, the fundus/corpus zone, and the pyloric zone, respectively, by the WGCNA algorithm [16]. The Cytoscape software [16] was used to visualize the gene co-expression networks [17], and Cytohubba was used to select hub genes [18] (Figure 2D and Appendix A). We also performed a GO and KEGG enrichment analysis of the purple (correlated with the cardiac zone), salmon (correlated with the pyloric zone), and brown (correlated with the fundus/corpus zone) modules (Appendix A).

We performed a survival analysis of the genes from the co-expression networks of the three zones on GC patients in the TCGA target GTEx and GSE66229 datasets (GSE15459 did not have information on the three zones). The results indicated that the top five genes (APOA4, MS4A10, SLC28A1, AQP10, and APOB) from the cardiac zone co-expression network correlated most significantly with the outcomes of GC patients in the cardiac zone compared to GC patients of the other two zones (Appendix A). The same was true for patients with GC in the fundus/corpus zone. The top five genes (VCAN, COL1A2, FAP, PODNL1, and SULF1) from the fundus/corpus zone co-expression network also displayed the vastest correlation with the fundus/corpus zone GC patients compared to GC patients of the other two zones (Appendix A). Meanwhile, we a performed binary classification using the logistic regression classifier (LR) and random forest classifier (RF) with the hub genes (APOA4, MS4A10, SLC28A1, AQP10, and APOB) of GC in the cardiac zone as the feature genes to predict whether patients have GC, and we achieved the highest AUC in patients with GC of the cardiac zone compared to the other two zones in the GSE66229 datasets (Appendix A). The same was true for the hub genes (VCAN, COL1A2, FAP, PODNL1, and SULF1) of GC patients in the fundus/corpus zone (Appendix A). These results further indicated that discriminating among different zones of GC is of great importance.

### 3.2. Construction and Cross-Validation of the PRGS Model in Gastric Cancer Cohorts

To further determine the prognostic genes related to the three zones of GC, we continued to generate predictive genes using the TCGA target GTEx and cross-validated these genes with two independent datasets (GSE66229 and GSE15459). Based on the expression profiles of 25 genes correlated with GC in the cardiac zone, the fundus/corpus zone, and the pyloric zone, a LASSO–Cox regression analysis [19] generated the predictive genes. This process was repeated 1000 times with the glmnet R package to ensure the stability of the gene [28]. In the LASSO regression, the partial likelihood of deviance reached the minimum value to obtain the optimal λ (Appendix A). Six, four, and three genes from GC in the cardiac zone, the fundus/corpus zone, and the pyloric zone, respectively, with positive LASSO coefficients were subjected to the log-rank test and the Kaplan–Meier curve (Figure 3A), which identified a final set of four genes. There were four genes with the maximum coefficient of GC in the three zones, including APOB (*p*-value = 0.028, FDR = 0.0042) correlated with GC in the cardiac zone, VCAN (*p*-value = 9.2 × 10^−25^, FDR = 0.0004) correlated with GC in the fundus/corpus zone, and ABCA6 (*p*-value = 0.0028, FDR = 0.0042) as well as CTSF (*p*-value = 0.023, FDR = 0.023) correlated with GC in the pyloric zone. For each of these four genes, the increased expression level was vastly associated with a worse OS for gastric cancer patients (Figure 3A). Besides, these four genes (APOB, VCAN, ABCA6, and CTSF) are included in the DEG lists in the GC data of respective zones in dataset GSE66229.

We next depicted the PRGS expression level at a single-cell resolution. We analyzed the scRNA-seq data derived from 29 gastric cancer and 11 normal gastric tissues [29]. After the quantity control and removal batch effect, we obtained a total of 200,954 cells majorly comprising lymphoid cells (CD8A and KLRD1 positive), plasma (TNFRSF17 positive), epithelial cells (CDH1 positive), macrophages (CD163 positive), fibroblasts (FN1 and LUM positive), B cells (MS4A1 positive), mast cells (KIT positive), and pericytes (NOTCH3 positive) (Appendix A). As shown in Appendix A, VCAN was expressed in macrophages and fibroblasts; ABCA6 and CTSF were mainly expressed in fibroblasts; and APOB was expressed in epithelial cells (Appendix A). We also observed that GC samples in different Lauran classifications (intestinal, diffuse, and mixed types) exhibited higher PRGS scores than normal samples (Appendix A). We computed global PRGS scores for all cell types and found that fibroblasts had the highest PRGS scores compared to other cell types (Appendix A).

We also identified the chromatin accessibility of these four genes; thus, we analyzed the relative enriched proportions of coding regions, intergenic regions, introns, exons, and upstream and downstream regions (Appendix A) with ATAC-seq data from TCGA. The peak annotation demonstrated that the peaks of these genes were more likely to be located in promotor regions (Appendix A). We also performed motif enrichment and calculated potential regulatory TFs within the 200 bp range of gene loci based on genes from the PRGS model with HOMER (Appendix A). A KEGG analysis of the peaks was also performed (Appendix A).

The risk score for each patient was then calculated using the expression matrix of these four genes (APOB, VCAN, ABCA6, and CTSF) weighted by their regression coefficients in the Cox model (Appendix A). All patients were divided into high- and low-PRGS groups by the survminer package [30]. As can be seen from Figure 3B, the overall survival (OS) was significantly lower in the high-PRGS group relative to the low-PRGS group in the TCGA target GTEx training dataset and the two validation datasets (all with *p* < 0.05) (Figure 3B and Appendix A). In Appendix A, we see the distribution of the PRGS scores for the patient group, as well as the relationship between the PRGS and survival time. The cut-off for the high- and low-PRGS groups was 50%.

We also evaluated the PRGS in Lauren and WHO histotypes. The results shown in Appendix A also indicate that the overall survival (OS) of all classifications was significantly lower in the high-PRGS group relative to the low-PRGS group in the TCGA target GTEx training dataset and the two validation datasets (all with *p* < 0.05) (The GSE66229 and GSE15459 only provide the information of Lauran classification). Then, we assessed the OS status in different TNM stage groups. As shown in Appendix A, the high-PRGS group was tightly correlated with a worse OS status in late stages in TCGA target GTEx (T3, T4, N2, and N3) and in GSE66229 (T2, T3, N1, N2, and N3). The high-PRGS group was vastly associated with a worse OS status in the M0 stage and most of the patients in these datasets were in the M0 stage (Appendix A).

We noticed that patients with a high PRGS were distributed in all different stages and OS statuses (Appendix A), indicating that our PRGS model is prognosis-specific and capable of assessing the risk of GC patients regardless of stage and status, though clinically, it would be potentially used at later stages for precision.

To investigate whether the prognostic value of the PRGS based on the four genes was an independent risk factor, a multivariate Cox regression analysis was performed; the results indicated that the PRGS was notably associated with OS compared with other clinical characteristics (age, sex, clinical stages, and T, N, and M stages), thus validating that the PRGS is robust in independently predicting the GC prognosis (Figure 3C). Similarly, the PRGS was still an independent risk factor for OS in the validation datasets (GSE66229 and GSE15459) (Figure 3D). We further explored the performance of the PRGS with other characters and found that the performance of the PRGS was better than that of other factors, including gender (whether male or female), age, pathological stages (T1~4, N0~3, M0~M1), and clinical stages (stage I~IV) in TCGA target GTEx, GSE66229, and GSE15459 (Appendix A).

The discrimination of the PRGS was measured by an ROC analysis, with 1-, 3-, and 5-year AUCs of 0.601, 0.663, and 0.717 in TCGA target GTEx; 0.607, 0.709, and 0.707 in GSE66229; and 0.657, 0.674, and 0.711 in GSE15459 (Figure 4A and Appendix A). The C-index (95% confidence interval) of the PRGS was the highest compared with other factors (gender; age; T, N, and M; clinical stage) in the TCGA target GTEx cohorts (Appendix A).

### 3.3. PRGS Are Significantly Related to Clinical Outcomes

To further validate the performance of our PRGS model, we conducted multiple analyses to evaluate the robustness of these four prognostic genes in the aforementioned independent datasets (GSE66229 and GSE15459). First, through consensus clustering based on the four prognostic genes, we applied a consensus cluster analysis to all GC samples, resulting in the division of the samples into k clusters (k = 2–9). The cumulative distribution function (CDF) curves of the consensus score matrix and proportion of ambiguous clustering (PAC) statistic indicated that the optimal number was obtained when k = 2 [35]. We classified GC patients in GSE66229 and GSE15459 into Cluster 1 and Cluster 2 (Figure 4B,C and Appendix A), respectively. The Kaplan–Meier curve indicated significant OS differences between the two clusters via the log-rank test, and the OS of the patients in Cluster 1 was significantly worse than that of Cluster 2 (Figure 4D). As shown in Appendix A, the overall expression levels of the PRGS in Cluster 1 were higher than in Cluster 2 for both datasets. When k = 3 and 4, the Kaplan–Meier curve indicated that patients in Cluster 1 with the lower PRGS expression level had a vastly better OS than that of the other clusters (Appendix A). Thus, for these four genes, the increased expression level was associated with a worse survival rate for gastric cancer patients. The observed consistency suggests that the expression levels of these four genes are vastly correlated with the OS of patients.

There are many prognostic gene expression signatures that have been developed based on bioinformatics methods. To compare the performance of the PRGS with other signatures, we selected five cancer stem cell-related feature genes in GCscore risk models (FANCA, DUSP3, HIST1H3B, CLNS1A, and FANCC) [6], three common clinical GC biomarkers (CEACAM1, CEACAM5, and CEACAM6) [21], seven immune-related signatures (TGFB1, NOX4, F2R, TLR7, CIITA, RBP5, and KIR3DL3) [31], two cadherin gene signatures (CDH2 and CDH6) [32], six metastasis-related gene signatures (TMEM132, PCOLCE, UPK1B, PM20D1, FLJ35024, and SLITRK2) [33], eight methylation-based gene signatures (TREM2, RAI14, NRP1, YAP1, MATN3, PCSK5, INHBA, and MICAL2) [34], and three hypoxia-immune-based gene signatures (CXCR6, PPP1R14A, and TAGLN) [36] as features to classify gastric cancer patients with different machine-learning (ML) classifiers. We utilized the logistic regression (LR) classifier and random forest (RF) classifier ML models to predict whether patients had GC in GSE66229 (GSE15459 did not have normal samples). We performed binary classification with them at the same time. The results indicated that the PRGS had a low sensitivity to different classifiers while it had the highest and most robust AUC (Figure 4E and Appendix A). Additionally, when performing classification tasks on the patients in different clinical stages (stages I, II, III, and IV), the PRGS had the highest and most robust accuracy (Figure 4F, Appendix A). Hence, the PRGS signature had optimized effects in classifying GC samples from control samples, which could serve as a potential feature for examining patients’ prognoses. Together, we believe that the expression matrix based on these four genes (APOB, VCAN, ABCA6, and CTSF) as features for screening GC samples could properly assist classifiers in distinguishing cancerous samples from normal samples, acquiring satisfactory precision from test datasets and predicting whether patients have gastric cancer in each stage accurately.

### 3.4. Experimental Validation of the PRGS in the Clinical Samples and Cell Lines

To examine the protein expression levels of the four genes (APOB, VCAN, ABCA6, and CTSF) in gastric cancer, we performed immunohistochemistry (IHC) on the pathological section of the samples from patients of different stages, including normal, paracancerous tissue, early gastric cancer (EGC), and advanced gastric cancer (AGC) (Appendix A). Compared with the samples from the normal tissues, the expression levels of the four proteins (APOB, VCAN, ABCA6, and CTSF) were much higher in EGC (Figure 5A–D). These four genes showed strong expression in almost all of the AGC samples and showed weak expressions in a portion of the paracancerous areas (Figure 5E–H). These results indicated a higher expression level for the four genes (APOB, VCAN, ABCA6, and CTSF) in GC compared to normal and paracancerous lesions. Furthermore, these proteins have already reached high expression levels in EGC samples compared to the matching normal samples.

To investigate how these four highly-expressed genes influence tumorigenesis, we further performed anti-phosphorylated histone 3 (pH3) immunostaining and flow cytometry-based cell cycle assays (Figure 6A–D). Consistent with the results in human cancer tissues, all four genes were highly expressed in GES-1 and BGC803 gastric cancer cells (Figure 6E,F). Knocking down any of these four genes independently led to reduced phosphorylated histone 3, a well-established biomarker [37] of cell proliferation (Figure 6A,B). Consistently, the cell cycles of GC cells shifted from the S/G2/M to the G1 state upon the knockdown of APOB, VCAN, ABCA6, and CTSF, as demonstrated in flow cytometric analyses of the cell cycle with propidium iodide staining (Figure 6C,D). Note that the knockdown efficacy of the siRNAs used in our experiments were validated by quantitative real-time PCR (Figure 6E,F). These data unraveled that APOB, VCAN, ABCA6, and CTSF play important roles in the incidence of GC.

We detected the gene expression of the PRGS using an RT-PCR assay in a clinical cohort that included 19 normal gastric samples, 18 EGC patients, and 8 AGC patients by conducting qRT-PCR experiments (Figure 6G,H). The result indicated that the expression level in GC patients was significantly higher than in normal samples. These assays supported that our PRGS model was robust.

### 3.5. The Immune Cell Infiltration between the High- and Low-PRGS Patients

Immune-infiltrating cells in the tumor microenvironment (TME) can modulate the tumor phenotype. We assessed tumor purity as well as stromal and immune scores using the ESTIMATE algorithm in TCGA target GTEx samples [23]. ESTIMATE generates three types of scores, namely stromal scores indicating the presence of stroma, immune scores representing the infiltration of immune cells, and an ESTIMATE score, which infers tumor purity. Samples with a low tumor purity show high ESTIMATE scores [23]. The results demonstrated that stromal and immune cells significantly increased along with malignancy progression (from stages I to IV). The ESTIMATE score also increased from stage I to stage IV, whereas tumor purity decreased in higher grades (Figure 7A and Appendix A) in accordance with previous studies [38], which illustrated that lower tumor purity correlates with severer malignancy.

As expected, a high abundance of stromal cells and immune cells and a low tumor purity were shown in the high-PRGS group (Figure 7A). Hence, the high-PRGS group was positively correlated with malignancy. To further chart the underlying immune cells, we implemented the CIBERSORT algorithm to infer the differential abundance between the high- and low-PRGS patients [24]. ]. The high-PRGS group had a very high level of naïve B cells (*p* = 0.00014), monocytes (*p* = 0.0312), M2 macrophages (*p* = 1.6 × 10^−5^), and resting mast cells (*p* = 5.6 × 10^−5^) (Figure 7B and Appendix A), whereas T follicular helper cells (*p* = 3.9 × 10^−7^), Treg (*p* = 2.1 × 10^−6^), resting NK cells (*p* = 0.00716), and activated mast cells (*p* = 0.03602) exhibited a consistent negative correlation with the low-PRGS group. Then, we determined whether the infiltrating immune cells mentioned above could be associated with patient survival. In agreement with previous studies indicating the promoting roles of M2 macrophage cells in tumor progression [39] and naïve B cells when they differentiate into Breg cells in the tumor microenvironment, participating in tumor metastasis [40], we discovered that the naïve B cells and M2 macrophage cells appeared to be associated with poor survival (Figure 7C,D), consistent with the PRGS groups. In addition, the regulatory T cells (Treg), T follicular helper (Tfh) cells, and M0 macrophage cells showed a negative correlation with patient survival (Figure 7E,F), which accords with the low-PRGS group, where there were more Treg and Tfh cells than in the high-PRGS group. Together, our data indicate that the high- PRGS group was characterized by a TME with a high immune cell infiltration and a low tumor purity.

### 3.6. Mutation Status in GC Patients in the High- and Low-PRGS Groups

To investigate PRGS-related mechanisms in GC, we also analyzed somatic mutations. When comparing the mutant frequency between samples of the low- and high-PRGS groups, we detected more mutations in the low-PRGS group than the high-PRGS group, indicating that more mutations led to a lower PRGS and GC risk (Figure 7G). Past studies have revealed sophisticated correlations between mutations and tumor prognoses, e.g., TP53 mutations have significant associations with poor outcomes in kidney renal clear cell carcinoma, head and neck squamous cell carcinoma, and acute myeloid leukemia, as well as improved outcomes in ovarian serous cystadenocarcinoma [41]. Besides, IDH1 and MUC16 mutations are associated with an improved prognosis in gastric cancer [41,42,43,44,45]. All of the top 20 frequent mutations were significantly enriched in cases with patients in both groups (Figure 7G). As a reference, we utilized the GCscore [6] as feature genes (FANCA, DUSP3, HIST1H3B, CLNS1A, and FANCC) to classify GC patients from the TCGA dataset into high- and low-risk groups (Appendix A). The top 20 genes with the highest mutation frequencies also had relatively higher mutation frequencies in the low-risk group than in the high-risk group (Appendix A). This result exhibited a similar tendency as our PRGS. We also found that MUC16 (*p* = 0.051), CSMD1 (*p* = 0.054), and FAT4 (*p* = 0.033) mutants had better outcomes than wild type (WT), while TP53 and TTN, genes with the highest mutation frequencies, did not show differences in the outcomes between their mutants and the WT (Appendix A). Frequent mutations in TNN, TP53, and MUC16 were significantly enriched in the high- and low-PRGS groups, which were within expectations according to previous reports [46].

Additionally, APOB portrayed a mutation percentage of 8% and 18% in patients of the high- and low-PRGS groups, respectively. The mutation percentage for VCAN was 6% and 10% in the high- and low-PRGS groups; for ABCA6, it was 2% and 3%; and for CTSF, it was 1% and 3% (Appendix A). Previous work [47] has reported APOB and VCAN as mutation driver genes.

Moreover, we observed significant co-occurrences among mutations of these genes. Among the top 20 genes with the highest mutation probabilities in gastric cancer, the majority of these gene mutations were co-occurring. The gene mutations in the low-PGRS group displayed more significant co-occurrences than those in the high-PRGS group (Appendix A). There were also gene pairs that were mutually exclusive. For example, previous studies have reported that ARID1A mutant tumors display p53 pathway activation, and that ARID1A directly regulates TP53 target genes [48,49]; our analysis also displayed that in the high-PRGS group, TP53 and ARID1A mutations were significantly mutually exclusive. On the other hand, within the low-PRGS group, KMT2D and FAT4 were also significantly mutually exclusive along with the previous two. These results also indicate a possible pair of mutually exclusive mutations in patients within the low-PRGS group.

Somatic mutations were further explored on the basis of oncogenic signaling pathways. We summarized the gene expression levels and mutations of common oncogenic signaling pathways from patients of the high-PRGS group and low-PRGS group. For instance, disturbing the hippo pathway promoted GC proliferation and metastasis [50], and dysregulation of the MAPK pathway promoted cell metabolism, proliferation, apoptosis, and migration [51]. Our results also indicated that both the hippo and MAPK pathways had high mutation percentages for the samples from the high-PRGS and low-PRGS groups (Appendix A). We discovered that for the most investigated cancer pathways, the patients from the high-PRGS group had higher pathway enrichment scores than the patients in the low-PRGS group (Appendix A). Yet, the percentage of patients with mutations was lower in the high-PRGS group than in the low-PRGS group, which might have arisen from other molecular mechanisms.

## 4. Discussion

To date, there have rarely been effective prognostic genes identified for detecting and predicting gastric cancer prognoses. There is a noticeable difference among GC in the cardiac zone, the fundus/corpus zone, and the pyloric zone. However, researchers might neglect some significant genes when considering mixed data. Hence, we constructed co-expression networks of GC in these three zones separately to select prognostic genes and build the PRGS. Different treatment options mean that patients need better-individualized evaluations when implementing clinical decisions. These can be used as reliable biomarkers for the diagnosis of “high-risk” GC patients.

In this study, WGCNA and Cytoscape were applied to identify the gene co-expression networks of GC in the cardiac zone, the fundus/corpus zone, and the pyloric zone. With the expression profiles of these genes in TCGA target GTEx and two independent datasets, the LASSO–Cox regression model was applied to develop a prognostic gene signature. The prognostic analysis demonstrated that this PRGS was a deleterious indicator of the OS. Besides, the PRGS demonstrated a high accuracy and consistent performance for TCGA target GTEx. We also performed cross-validation using two independent GEO datasets (GSE66229 and GSE15459), which indicated great potential for the clinical application of the PRGS.

The common tools for evaluating clinical outcomes and making therapeutic schedules include T, N, M, and the clinical stage [4]. Remarkably, our signature worked independently of these factors and had a vastly superior performance in predicting prognoses. Besides, we took the PRGS as features to classify GC and normal samples with machine-learning classifiers. All of the classification accuracy based on the PRGS signature as features were highest with independent datasets (GSE66229) and for several clinical stages (stages I, II, III, and IV).

There are a number of prognostic gene signatures based on characteristic genes, such as stem cell-related gene signatures [6] and DNA methylation-related gene signatures [34]. These gene signatures overlook the differences among tumor locations. Although several of these signatures have been developed, few have been implemented in clinical experiments and even fewer have undergone rigorous validation. We compared the PRGS with these signatures to classify GC patients, and the PRGS had a better performance than the other signatures. To further confirm the clinical significance of the PRGS, we conducted a validation assay using qRT-PCR on 45 frozen gastric cancer tissues. The results further supported the validity of the PRGS as a clinical marker. Consequently, our signature has the potential to be a useful clinical tool for the prognosis determination of GC.

To further test the clinical explanation of the PRGS, the experimental validation was based on the IHC results from different clinical stages of independent GC patients, validating our prior findings and assessing their feasibility. This indicated that the protein expression level of the PRGS signature was significantly higher in GC patients. This conclusion was further experimentally validated in GES-1 and BGC803 gastric cancer cells, as the knockdown of these four genes led to cell growth inhibition by regulating the cell cycle. Hence, the PRGS signature could serve as a promising surrogate for assessing the prognosis of GC in clinical settings.

Pathologists generally determine tumor purity by visual evaluation, which is affected by the sensitivity of histopathology, interobserver bias, and variability in accuracy [52]. Our results show that the high-PRGS-group patients had a lower tumor purity and higher levels of immune and stromal cell infiltration compared with patients in the low-PRGS group. Previous studies have reported that tumor cells can dominate the microenvironment [52], which has given rise to the hypothesis that malignant GC recruits abundant surrounding cells and subjugates them to compose a protective shield. Therefore, a lower tumor purity and correlated cellular heterogeneity may contribute to a worse prognosis for GC. In our study, we found that the PRGS was positively correlated with several infiltration cells, such as M2 macrophages and naïve B cells, which are positively correlated with the prognosis of GC patients; furthermore, the PRGS was negatively correlated with follicular helper T cells, M0 macrophages, and regulatory T cells, which are negatively correlated with the prognosis of GC patients.

We found that GC patients in the low-PRGS group had a higher rate of oncogenic mutations. It should be noted that the association of gene mutations with cancer outcomes is sophisticated. Take MUC16 and IDH1 as examples: past GC studies have illustrated that MUC16 mutations could activate the p53 pathway and DNA repair pathway, which are all tumor suppressor pathways [42,43]; thus, improved outcomes of GC might be expected from mutated MUC16 [34,36]. Mutations in IDH1, as a tumor suppressor in human glioma cells through the negative regulation of Wnt/β-catenin signaling, improves survival conditions [44,45]. In addition, mutations in the top 20 genes had a high frequency of co-mutations. However, further investigations are needed for a deep understanding of the mechanisms of mutations in GC.

## 5. Conclusions

In a word, based on a series of bioinformatics, machine-learning-based algorithms, and experimental validation, we developed a powerful and robust signature for assessing the prognosis of GC patients. This PRGS model may be a promising tool for screening and monitoring individual GC patients.

## Figures and Tables

**Figure 1 cancers-15-01610-f001:**
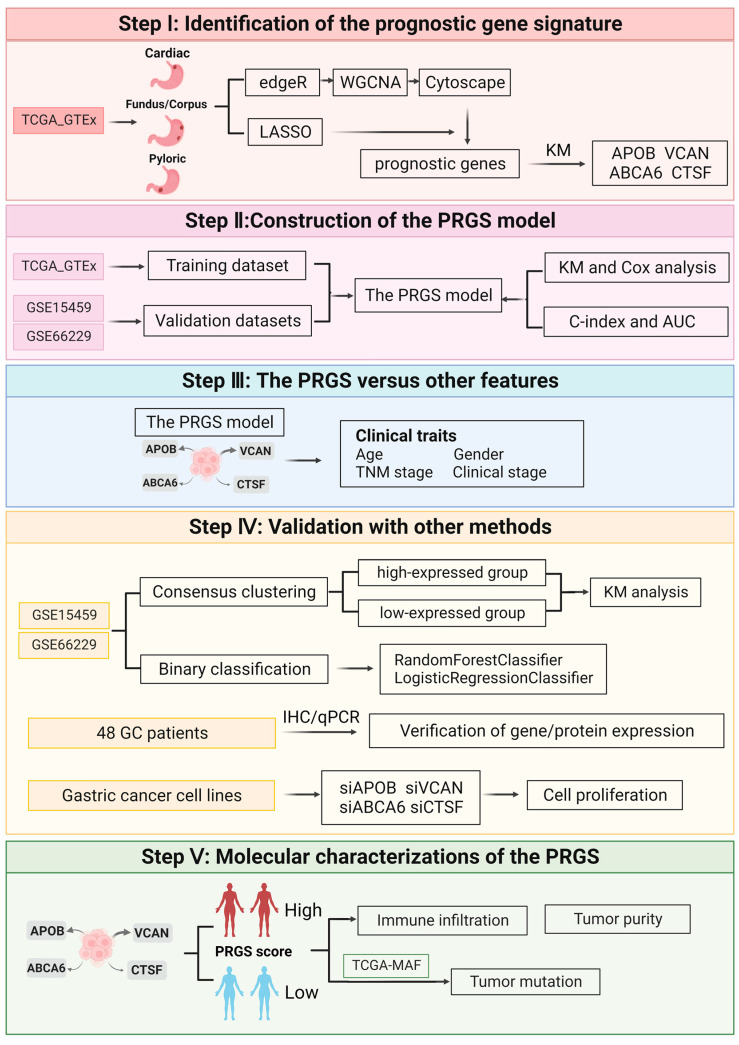
The overall design of this study.

**Figure 2 cancers-15-01610-f002:**
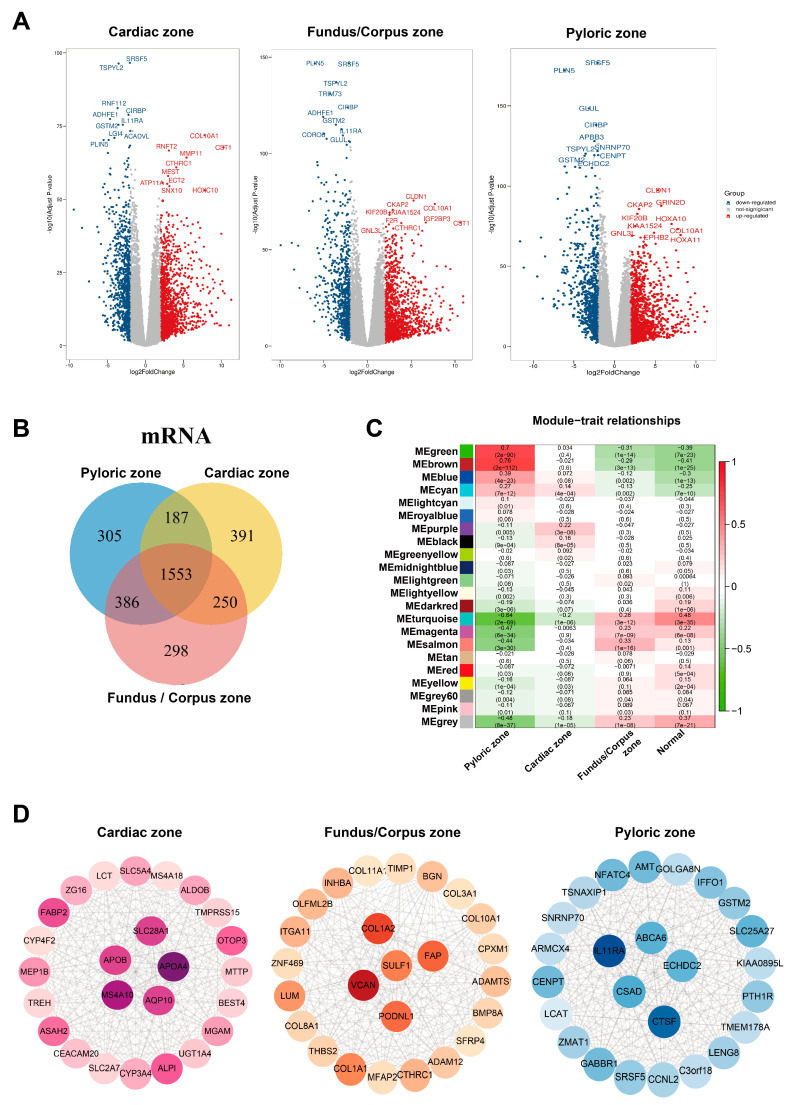
Identified co-expression networks of the cardiac zone, the fundus/corpus zone, and the pyloric zone. (**A**) Volcano plots of DEGs in the cardiac zone, the fundus/corpus zone, and the pyloric zone. Red and blue spots represent significant up- and down-regulated RNAs, respectively. The remark for the gene symbols represents the significant up- and down-regulated RNAs. (**B**) Venn diagram of the DEGs in the cardiac zone, the fundus/corpus zone, and the pyloric zone. Different colors indicate GC in different zones. The numbers represent the common DEGs among GC in different zones. (**C**) Correlation analysis between module eigengenes and the three zones. (**D**) Co-expression networks of the cardiac zone, the fundus/corpus zone, and the pyloric zone.

**Figure 3 cancers-15-01610-f003:**
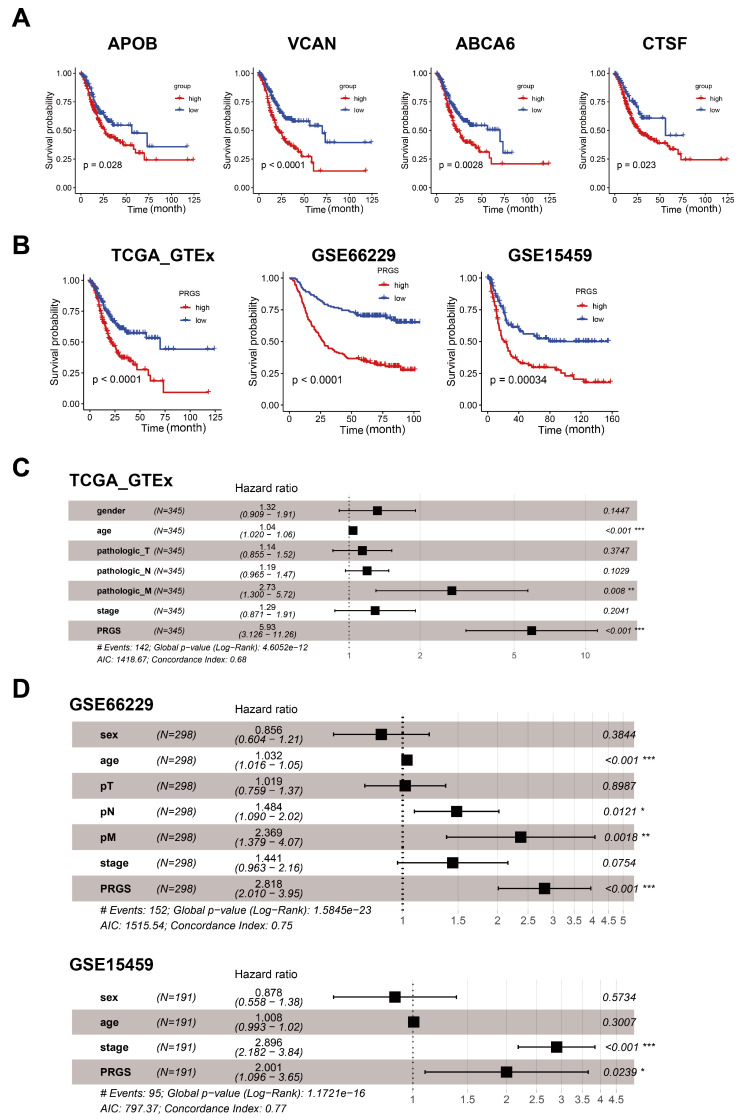
Cox and survival analysis of the PRGS model. (**A**) Kaplan–Meier curves of overall survival (OS) according to the PRGS in TCGA target GTEx. (**B**) Kaplan–Meier curves of OS according to the PRGS in TCGA target GTEx (log-rank test: *p* = 7.75 × 10^−5^); GSE66229 (log-rank test: *p* = 2.11 × 10^−11^); and GSE15459 (log-rank test: *p* = 0.00034). (**C**) Multivariate Cox regression of PRGS regarding OS in TCGA target GTEx (n = 345). (**D**) Multivariate Cox regression of PRGS regarding OS in GSE66229 (n = 298) and GSE15459 (n = 191). Statistic tests: two-sided Wald test. Data are presented as hazard ratio (HR) ± 95% confidence interval (CI). * *p* < 0.05. ** *p* < 0.01. *** *p* < 0.001.

**Figure 4 cancers-15-01610-f004:**
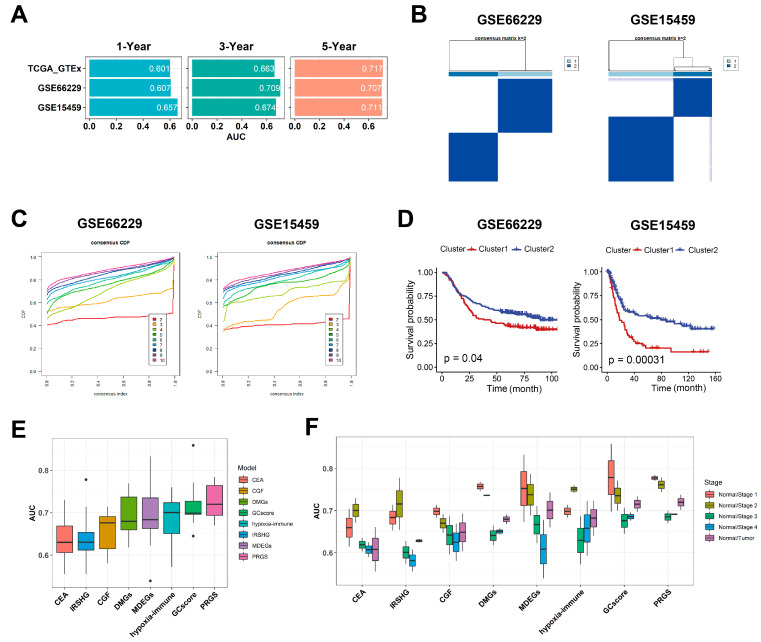
Evaluation of the PRGS signature. (**A**) A time-dependent ROC analysis for predicting OS at 1, 3, and 5 years in TCGA target GTEx, GSE66229, and GSE15459. (**B**) The consensus score matrix of all samples in GSE66229 and GSE15459 when k = 2. A higher consensus score between two clusters indicates they are more likely to be grouped into the same cluster in different iterations. (**C**) The CDF curves of the consensus matrix for each k. (**D**) Kaplan–Meier analysis showed that patients in Cluster 1 exhibited a worse OS in both the GSE66229 (*p* = 0.04) and GSE15459 (*p* = 0.00031) cohorts. (**E**) Box plot of AUC of two classifiers (LR, logistic regression; RF, random forest) with eight gene signatures as features in GSE66229. (**F**) Box plot of AUC of CEA, GCscore, and PRGS in clinical stages I to IV. CEA [21] is three common clinical GC biomarkers, including CEACAM1, CEACAM5, and CEACAM6; GCscore [6] includes FANCA, DUSP3, HIST1H3B, CLNS1A, and FANCC; PRGS includes APOB, VCAN, ABCA6, and CTSF; IRSHG includes seven immune-related signatures (TGFB1, NOX4, F2R, TLR7, CIITA, RBP5, and KIR3DL3) [21]; CGF includes two cadherin gene signatures, CDH2 and CDH6 [31]; DMGs include six metastasis-related gene signatures (TMEM132, PCOLCE, UPK1B, PM20D1, FLJ35024, and SLITRK2) [32]; MDEGs include eight methylation-based gene signatures (TREM2, RAI14, NRP1, YAP1, MATN3, PCSK5, INHBA, and MICAL2) [33]; and hypoxia-immune-based gene signature includes CXCR6, PPP1R14A, and TAGLN [34].

**Figure 5 cancers-15-01610-f005:**
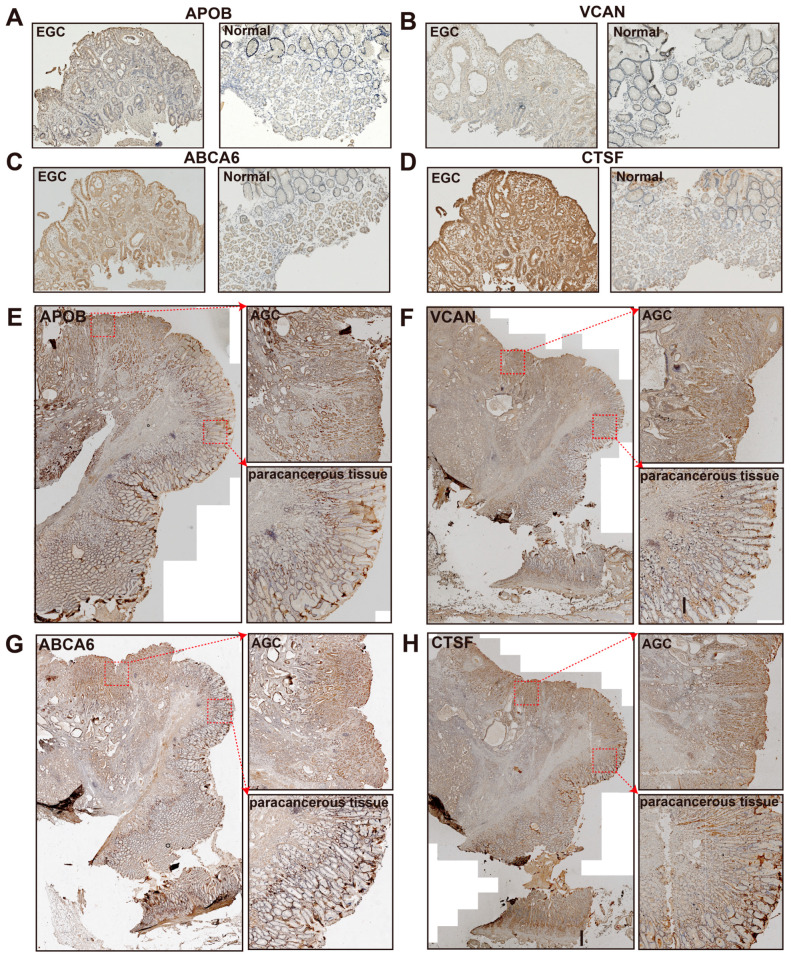
The results of immunohistochemistry (IHC). (**A**–**D**) IHC staining of APOB (**A**), VCAN (**B**), ABCA6 (**C**), and CTSF (**D**) in an early gastric cancer sample (EGC, left) and a normal sample (right). (**E**–**H**) IHC staining of APOB (**E**), VCAN (**F**), ABCA6 (**G**), and CTSF (**H**) in advanced gastric cancer (AGC). The pathological section of AGC shows the overall site (left), tumor region (AGC, upper right), and paracancerous tissue region (lower left).

**Figure 6 cancers-15-01610-f006:**
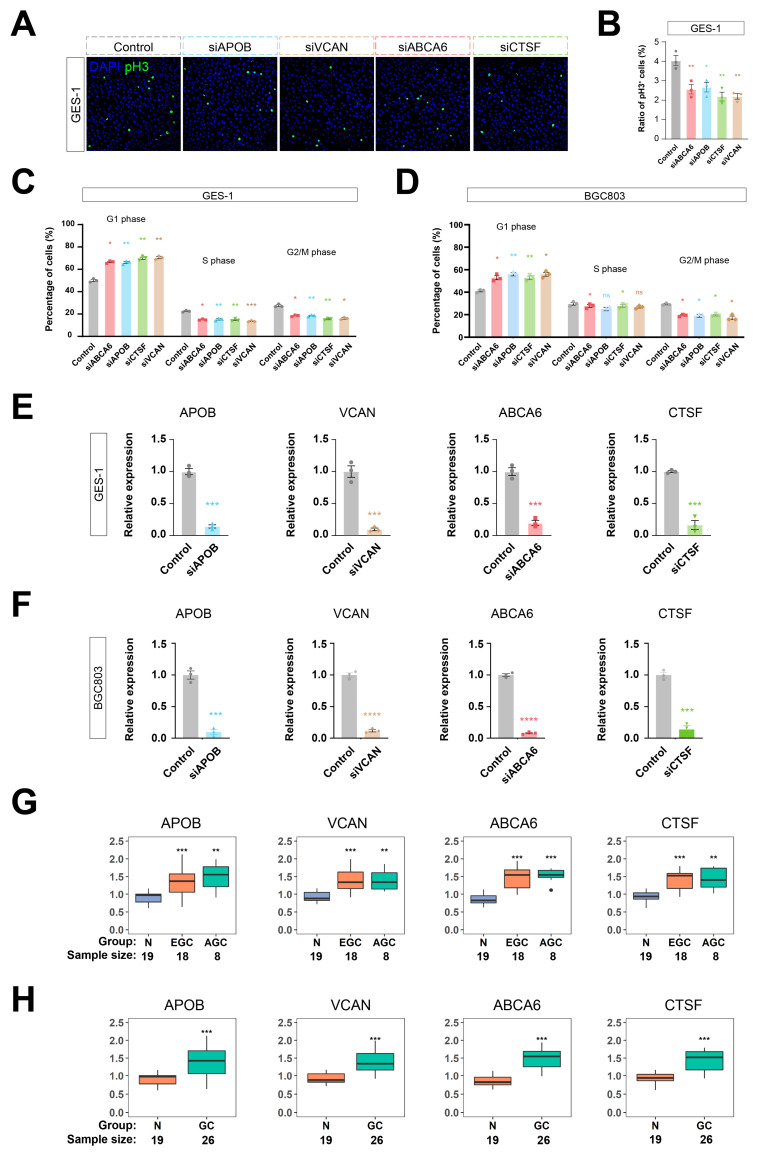
Knockdown of PRGS inhibited proliferation of gastric cancer cells. (**A**–**D**) Knockdown of any one of APOB, VCAN, ABCA6, or CTSF by siRNA inhibited cell proliferation of GES-1 and BGC803 gastric cancer cells. Shown are the anti-phosphorylated histone 3 (pH3) immunostaining (**A**,**B**) and flow cytometry-based cell cycle assays (**C**,**D**) of the APOB, VCAN, ABCA6, or CTSF siRNA treatment. For (**A**), scale bar is 10 μm. For (**B**–**F**), n = 3. One-way ANOVA and Tukey’s multiple comparison tests were used. In all figures, standard errors of the mean were represented. NS is not significant. * *p* < 0.05. ** *p* < 0.01. *** *p* < 0.001, **** *p* < 0.0001. (**G**) Box plot of PRGS expression level in normal, early gastric cancer, and advanced gastric cancer patients. (**H**) Box plot of PRGS expression level in normal and gastric cancer patients. N means normal gastric tissues; EGC means early gastric cancer tissues; AGC means advanced gastric cancer tissues; and GC means gastric cancer tissues.

**Figure 7 cancers-15-01610-f007:**
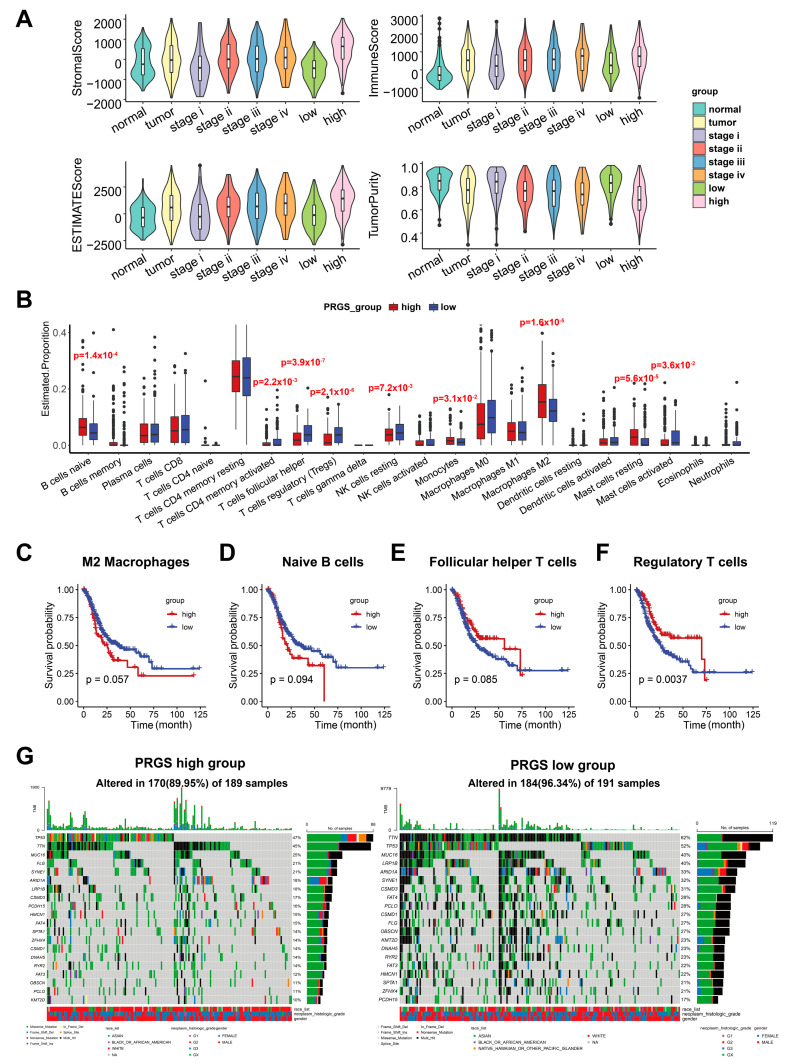
The landscape of immune infiltrations and tumor mutation status in the low- and high-PRGS groups. (**A**) Distinct distribution of stromal score (upper left), immune score (upper right), ESTIMATE score (lower left), and tumor purity (lower right) among GC patients in TCGA target GTEx. (**B**) The immune-infiltrating cells in the TME were determined based on the CIBERSORT algorithm in the TCGA target GTEx. (**C**–**F**) Kaplan–Meier curves of OS according to the M2 macrophages (**C**), naïve B cells (**D**), follicular helper T cells (**E**), and regulatory T cells (**F**) in TCGA target GTEx. (**G**) OncoPlot of significantly mutated genes in high- (left) and low- (right) PRGS groups. The mutation types with their frequencies are presented.

## Data Availability

The datasets that support the findings of this study are available from the Cancer Genome Atlas TARGET and Genotype Tissue Expression project datasets (TCGA target GTEx, primary_site = stomach) and the NCBI Gene Expression Omnibus (GEO) under accession numbers GSE66229 and GSE15459.

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
