# Peer review of "Development and Experimental Validation of a Novel Prognostic Signature for Gastric Cancer"

_cancers, 2023, doi:10.3390/cancers15051610_

Round 1
Reviewer 1 Report
In the present study, a prognostic model for gastric cancer (PRGS) using gene signature was developed. Although few patient-, treatment- and pathological-related variables, commonly used in clinical practice, were included in the model, the study is interesting. I have some comments:
- Lauren and WHO histological classifications were not considered in the study. Recently, increasing attention is paid in literature above all to diffuse/poorly-cohesive types, including their genetic phenotype and associated microenvironment. It may be interesting to evaluate gene signature in different histotypes. Furthermore, Lauren/WHO histotypes should be considered in the prognostic model as potential prognostic variables.
- - Survival curves according to PRGS score in different TNM stage groups should be provided.
- - Please consider in the Discussion other clinical prognostic scores for gastric cancer, and what PRGS could add in clinical practice.
- - Please provide the reference list (we didn’t find it in the paper).
Author Response
Please see the attachment, Thank you!

Reviewer 2 Report
In this manuscript, the authors established a novel prognostic signature for gastric cancer. As we know, gastric cancer is a malignant tumor with high morbidity and mortality. Therefore, accurate recognition of prognostic molecular markers is the key to improving efficacy and prognosis. My suggestions are as follows,
Major:
1. Detailed information of data acquisition and processing is needed, for example, did the author remove the batch effect? How did they deal with multiple probes aimed at the same gene? Which type of RNA-seq data was downloaded? (Count, FPKM, or TPM)
2. The formula of PRGS score should be explained.
3. To verify the robust efficacy of this model, it is necessary to make comparison with other published models related to gastric cancer.
4. More bioinformatic analysis should be added, for example, the author should perform some single-cell analyses.
5. What’s the cut off value of PRGS score in each cohort?
Minor:
1. In consensus clustering, the results of k=3 and k=4 should be added in supplementary part.
2. Some texts is overlapped with the figures, please check it.
3. The manuscript should be edited by a naïve speaker.
Author Response
Please see the attachment, thank you so much for your guidance.
Round 2
Reviewer 2 Report
The authors have made perfect revisions according to muy suggestions. I suggest it might be better if the author could add the distribution of PRGS at the scRNA level.
Author Response
Please see the attachment, thank you so much for your suggestion.
